# Recent Advances in Cellulose-Based Biosensors for Medical Diagnosis

**DOI:** 10.3390/bios10060067

**Published:** 2020-06-17

**Authors:** Samir Kamel, Tawfik A. Khattab

**Affiliations:** 1Cellulose and Paper Department, National Research Centre, Cairo 12622, Egypt; samirki@yahoo.com; 2Dyeing, Printing and Auxiliaries Department, National Research Centre, Cairo 12622, Egypt

**Keywords:** cellulose, optical, electrochemical, bio-molecules, diagnostic tools

## Abstract

Cellulose has attracted much interest, particularly in medical applications such as advanced biosensing devices. Cellulose could provide biosensors with enhanced biocompatibility, biodegradability and non-toxicity, which could be useful for biosensors. Thus, they play a significant role in environmental monitoring, medical diagnostic tools, forensic science, and foodstuff processing safety applications. This review summarizes the recent developments in cellulose-based biosensors targeting the molecular design principles toward medical detection purposes. The recognition/detection mechanisms of cellulose-based biosensors demonstrate two major classes of measurable signal generation, including optical and electrochemical cellulosic biosensors. As a result of their simplicity, high sensitivity, and low cost, cellulose-based optical biosensors are particularly of great interest for including label-free and label-driven (fluorescent and colorimetric) biosensors. There have been numerous types of cellulose substrates employed in biosensors, including several cellulose derivatives, nano-cellulose, bacterial cellulose, paper, gauzes, and hydrogels. These kinds of cellulose-based biosensors were discussed according to their preparation procedures and detection principle. Cellulose and its derivatives with their distinctive chemical structure have demonstrated to be versatile materials, affording a high-quality platform for accomplishing the immobilization process of biologically active molecules into biosensors. Cellulose-based biosensors exhibit a variety of desirable characteristics, such as sensitivity, accuracy, convenience, quick response, and low-cost. For instance, cellulose paper-based biosensors are characterized as being low-cost and easy to operate, while nano-cellulose biosensors are characterized as having a good dispersion, high absorbance capacity, and large surface area. Cellulose and its derivatives have been promising materials in biosensors which could be employed to monitor various bio-molecules, such as urea, glucose, cell, amino acid, protein, lactate, hydroquinone, gene, and cholesterol. The future interest will focus on the design and construction of multifunctional, miniaturized, low-cost, environmentally friendly, and integrated biosensors. Thus, the production of cellulose-based biosensors is very important.

## 1. Introduction

Cellulose is an inexhaustible widespread biopolymer with an interesting structure and characteristics. It consists of glucose-based polymer chains as a major constituent of the plant cell-wall. Annually, plants naturally produce about 10^11^ tons of cellulose [1]. Besides its natural abundance, renewability, biocompatibility, and biodegradability, cellulose exhibits unique characteristics, such as transparency, dimensional stability, a high Young’s modulus, and a low thermal expansion coefficient and can easily be chemically modified [2,3,4,5]. Due to the hydrophilic nature of cellulose, it is not well-suited with the hydrophobic nature of some molecular sensors. Thus, additional chemical treatments should be applied. Cellulose derivatives have been used for a variety of applications, such as in the pharmaceutical industries, coatings, textiles, foodstuffs, immobilization of antibodies and proteins, optical films, laminates, and production of composites bearing both synthetic polymers and biopolymers [6,7,8,9,10,11]. In addition, the potential use of cellulose as a smart material has been investigated. The cellulose actuation mechanism was firstly reported by Kim et al. [12]. Upon applying an electric voltage to electrodes, cellulose was found to function as an actuator by generating a bending displacement. To enhance the cellulose-based actuator performance, a conductive polymer coating was applied to the cellulose substrate [13,14]. The use of a single and multi-walled carbon nanotube-cellulose hybrid-based actuator was found to enhance its performance in terms of force and actuation frequency [15]. Cellulose-based nanocomposites have been investigated for disposable chemical sensors, biosensors, and energy conversion devices [16,17,18,19,20,21,22]. The immobilization of a metal oxide onto the cellulose matrix made it suitable for utilization in the production of bioelectronics due to its acquired mechanical properties, chemical stability, photosensitivity, and conductivity [23].

The large surface area and porous structure of a fibrous cellulose substrate result in the rapid adsorption and diffusion of the analyte to the active detective sites through the mesh [24,25,26,27,28,29]. Thus, cellulose and its derivatives are characterized by improved sensitivity, accuracy, and rapid response. Cellulose strips can be employed as rigid or semi-rigid scaffolds in paper-based biosensors due to its large surface area to volume ratio and highly porous structure, which enables immobilizing analytes and reagents for future utilization toward the analysis of liquid or vapor samples. Thus, paper-based biosensors are cost-effective detection tools with the ability to monitor significant biomarkers of Parkinson’s disease [30] or those existing in body fluids, such as *α*-amylase [31]. Cellulosic paper biosensors are mainly composed of cellulose strips, and stimuli-responsive active sites usually characterized by their low cost, portability, and being disposable [32,33,34,35,36,37,38,39,40]. These distinctive advantages make cellulose sensing strips typical alternatives to other biosensors for a variety of analytes, such as hydrogen sulfate, deoxyribonucleic acid, and moisture [41,42,43]. Dopamine (3,4-dihydroxyphenethylamine) was employed as a stimuli-responsive active material, based on exonuclease III-mediated cycle amplification, in developing cellulose paper biosensors for the equipment-free and visual detection of a model transcription factor. Compared to other transcription factor biosensors, this biosensor was characterized as having a naked eye and equipment-free detection, low-cost, portability, and disposability [44]. These biosensors can be modified with nanomaterials, such as gold nanoparticles (AuNPs) or silver nanoparticles (AgNPs), to introduce a Plasmonic field, color change, or fluorescence quenching [45]. Additionally, the nanoparticles labeled with enzymes enable the multi-step chemical amplification of the biosensor analytical signal to achieve an ultra-enhanced sensitivity and better limit of detection [46].

In this review, we present recent methods on the preparation and modification of cellulose substrates as biosensors. A sensor can be defined as a tool that is able to measure a physical identity and translate it into a recognized signal. This signal can be monitored using an electronic device or an observer. A biosensor is a sensor device that can react with different biological components. In the last few decades, biosensing technology has been increased toward the development of bio-recognition advances, transducers, and signal processing. The “key-lock” biosensors function by recognizing the physico-chemical variations caused by a bio-receptor “key” interacting with an analyte “lock” [47]. Thus, biosensors can detect biomolecules, such as urea, estrogen, glucose, and cholesterol. The key element of a biosensor is the transducer, which causes the utilization of a physico-chemical change to be accompanied by a reaction. A sensor mediator is a material that can be employed to facilitate electron transfer between the electrode and the analyte and typically are coated on the electrode surface. Thus, the sensor response is highly affected by the chemical composition of the mediator [48]. There are several types of transducers, such as clays, metallic nanotubes, and nanoporous alumina membranes. In addition, there has been recently a growing interest in utilizing polymer-based hybrid composites as biosensors [49,50,51]. Polymer-based biosensors have received more attention owing to their efficiency in collecting a huge number of analyte molecules to be detected on their sensing surface. Within those biosensors, the recent findings of cellulose-based smart materials have paved the way to the utilization of cellulose as a potential candidate for biosensors. Cellulose-based biosensors are characterized by a number of desirable properties, such as sensitivity, rapid response, accuracy, and low cost. Thus, they play a significant role in environmental monitoring, medical diagnostic tools, forensic science, and foodstuff processing safety applications. The biocompatibility, biodegradability, and non-toxicity of cellulose make it an attractive material to be used in biosensors. Cellulose-based biosensors can be divided into two major categories, including cellulose-based optical and electrochemical biosensors. They can be applied during the sampling process without using sophisticated instrumentation. This review presents the recent developments and significant applications of cellulosic biosensors.

## 2. Classification of Biosensors

A biosensor is defined as a device that can convert (bio)chemical information into an electronic signal via a suitable transducer containing a certain molecular recognition structure. Thus, a biosensor can be described as an integrated receptor-transducer device with the ability to introduce precise and selective quantitative or semi-quantitative analytical data employing a biologically active component [52,53]. A biosensor is mainly made up of three elements, including a biologically active element immobilized on a convenient substrate such as cellulose, a transducer, and a signal processor, as shown in Figure 1. The biologically active element could be an enzyme, antibody, protein, whole cell, or DNA [54].

The biosensors can be classified according to the bio-receptor properties, such as enzymes, DNA, microbial whole cells, antibodies, and proteins or fragments. They can also classified depending on the physico-chemical properties of transducer, such as optical, electrochemical, calorimetric, thermal, or piezoelectrical. In the fabrication of biosensors, the major difficulty occurs during the immobilization of the biological active materials with the transducers onto physical matrices. This immobilization process could be carried out by direct incorporation with or without using a bifunctional agent, or upon using specialized membranes with or without the insertion of bifunctional agents. The development of a biosensor for detecting a certain analyte in a broad range of concentrations without interference from other materials depends mainly on the selection of a suitable bio-receptor, a suitable immobilization method, and a precise transducer and being wrapped in a portable form [55,56]. Compared to conventional analytical detection methods, biosensors are characterized by their low cost, rapid response, and portability, making them possible real time and in situ monitoring systems. Implantable biosensing devices can provide the continuous healthcare monitoring of metabolites, affording early warning tools of metabolic balances assisting in preventing and curing various disorders, such as obesity and diabetes [13]. Enzymes have been used as biological active materials employed in the development of biosensors owing to their specificity, fast response, high sensitivity, portability, robustness, reliability for testing, and low cost. Enzymes can monitor a certain analyte by measuring either the consumption or production of a specific substance, such as ammonia, oxygen, carbon dioxide, hydrogen peroxide, or hydrogen ions, and consequently transducers recognize the pollutant and correlate their occurrence in the substrates. Due to their poor stability in solution, enzymes need to be stabilized by incorporation into a stable and compatible matrix, which could be performed by cross-linking, physical adsorption, covalent bonding, or encapsulation [57,58]. The selection of the incorporation technique depends mainly on the nature of the biological element, the type of transducer, the analyte physico-chemical properties, and the biosensor operational conditions. Furthermore, it is necessary that the biological element demonstrates maximum biological activity in the integration matrix. The development of enzyme-based biosensors is in a good harmony with green chemistry concerns, because it is a clean process. However, the utilization of enzyme-based biosensors is limited under some specific situations due to some shortcomings, such as the limited lifetime of enzymes as well as their high sensitivity to environmental aspects such as pH and temperature. Thus, enzyme-free biosensing tools have been introduced due to their facile development, reproducibility, and stable properties. New functional electrodes and other nanoparticle-coated electrodes have been examined in the fabrication of enzyme-free biosensing devices [54].

## 3. Cellulose Functionalization

The demands for goods manufactured at low environmental safety risk from non-petroleum, renewable, and sustainable resources have been persistently increased by consumers and industry. Half of the biomass generated by photosynthesis of organisms, such as some bacteria, algae, and plants is composed of cellulose. Thus, cellulose is the highest naturally abundant material worldwide. Cellulose is naturally available in various forms of materials, such as wood and cotton [59,60]. Natural cellulose exhibits various characteristics, including biocompatibility, biodegradability, hydrophilicity, chirality, ability for wide chemical modifications to yield various cellulose derivatives, and the capacity to generate semi-crystalline fibrous morphologies. Due to these characteristics, cellulose has drawn much increased interest and promoted interdisciplinary research on cellulose-based materials and products. Its solubility depends on several aspects, particularly its molecular weight, origin source, and structure. Generally, polysaccharides have been known for their strong affinity for aggregation or their poor solubility owing to the creation of hydrogen bonds, which is one of the most significant aspects affecting its chemical and physical features. The solubility, crystalline properties, and reactivity of the hydroxyl functional groups on the cellulose polymer chains can be directly influenced by both intra- and intermolecular bonding patterns (Figure 2) [61,62,63].

Cellulose derivatives, such as carboxymethyl cellulose, cellulose acetate, and cellulose nitrate, are usually designed and tuned to introduce certain desired characteristics via chemical functionalization by affording different functional substituents on cellulose polymer chains and/or by changing the inherent hydrogen bonding network (Figure 3). The properties of the produced cellulose derivatives are largely correlated to the degree of substitution and the type of functional substituents used. The functional groups can avoid the spontaneous creation of H-bonds or even generate novel interactions amid the cellulosic polymer chains. Thus, advances in cellulose chemical modifications have been recently performed to accomplish novel methods for the manufacture of sustainable and functional cellulosic substrates. Modifying the surface of cellulose chemically is a traditional method to convert the polar hydroxyl functional substituents located at the surface into moieties with the ability to improve interactions with the immobilization matrix [59,60,61,62,63,64,65].

Thus, there are various new functionalized cellulose-based materials that have been introduced for biosensors. The high number of hydroxyl functional substituents on cellulose polymer chains makes it a useful solid material that can be functionalized, leading to new advanced applications. Due to the rigidity of the polymer chains, some cellulose-based derivatives can generate lyotropic or thermotropic liquid crystalline mesophases. Those cellulose-based liquid crystals, with a periodically and internally modulated refractive index, displayed various significant optical characteristics due to their photonic band structure, leading to a variety of applications such as energy storage devices, information displays, polarized light sources, and mimics of biological tissues [66,67,68,69,70,71].

## 4. Cellulose-Based Biosensors

Both cellulose and its derivatives exhibit a high biocompatibility, making them suitable substrates for the incorporation of biologically active substances. A typical carrier substrate for an enzyme should be stable and inert as well as resistant to mechanical changes, making the utilization of a cellulose matrix ideal for chemical covalent bonding as well as physical adsorption and immobilization [72,73]. In general, biosensors can be divided into optical, electrical, and mechanical, which involve a broad range of detection approaches, such as bioluminescent, fluorescent, colorimetric, quartz microbalance, electrochemical, conductometric, electronic, gravimetric, piezoelectric, pyroelectric, and acoustic techniques [74]. According to the previous studies, cellulose-based biosensors can be divided into two major categories depending on their bio-transducer type, including cellulose-based optical and electrochemical biosensors. The authors of this review developed simple, portable, fast and highly sensitive cellulose strips for the fluorescent and naked-eye colorimetric detection of ammonia in both vapor and aqueous phases under ambient conditions. The first sensor strip was fabricated by loading tricyanofuran hydrazone molecular switching probes to function as stimuli-responsive active sites onto cellulose/polyethylene terephthalate composites [75]. In another trial, they immobilized bromocresol purple into cross-linked carboxymethyl cellulose to obtain a vapochromic (vapor-responsive) xerogel for the real-time sensing of ammonia with a limit of detection at 9.0 × 10^−2^ ppb [76]. Additionally, a “turn-on” fluorescent platform was reported by incorporating a thin film of fluorescent chitosan nanoparticles onto cellulose strips for selective the sensing of ammonia and amine vapors [77]. Another derivative of the tricyanofuran hydrazone chemical probe was incorporated onto smart microfibrillated sponge-like cellulose aerogel to provide a simple and flexible colorimetric swab sensor for sweat monitoring, demonstrating color changes from yellow to orange and red and blue depending on the sweat biochemical variations [78,79].

### 4.1. Cellulose-Based Optical Biosensors

Optical biosensors are robust recognition and analytical tools that have various applications in different fields, such as biomedical, pharmaceuticals, healthcare, homeland security, battlefield, and environmental monitoring [80,81,82,83]. In general, there are two detection optical biosensing protocols, including label-driven (colorimetric or fluorescent) and label-free detection tools. In label-driven (colorimetric or fluorescent) detection tools, either the bio-recognition molecules or the target molecules are labeled with colorimetric or fluorescent tags, such as dye-based probes [84,85,86]. Either color change (colorimetric or chromic) due to a UV-Vis absorption intensity change or the presence/absence of emission (fluorescence) intensity proves the existence of a target molecule as well as the extent of interaction among the target molecule and the bioactive detective molecular systems. The label-driven detection tools are usually highly sensitive, with a limit of detection below the single molecule [87,88]. However, they suffer from hard labeling procedures. In addition, quantitative analysis is a challenge owing to the label signal bias, as the amount of chromophores (in the colorimetric case) or fluorophores (in the fluorescence case) on each molecule cannot be accurately controlled. In label-free biosensors, the target molecules are not labeled, introducing a relatively simple and inexpensive technique. This allows for kinetic and quantitative measures of the molecular interactions. Moreover, some label-free recognition mechanisms depend on refractive index changes stimulated by the molecular interactions correlated to the surface density or concentration of the tested sample rather than the total sample mass. Hence, the recognition signal does not depend on the volume of the sample [89,90]. This feature is especially attractive upon applying a very small (femto- to nanoliter) volume, making it advantageous over label-driven biosensors, whose signals usually rely on the analyte concentration, the recognition surface, or the recognition volume. Despite these differences between label-driven and label-free biosensors, both techniques are being broadly employed in optical sensors and present vital and complementary information concerning interactions between bio-molecules, making optical biosensors more versatile compared to other classes of biosensors [87,88,89,90]. Colorimetric and fluorescent optical biosensors are very simple technique allowing naked-eye detection, which makes them particularly attractive biosensing tools compared to the electrochemical biosensors. Thus, optical biosensors are recognized as easy to function, low cost, reusable and efficient analytical approaches employed to monitor the primary hazards or changes in the surrounding environment [42,87,88].

#### 4.1.1. Label-Free Optical Biosensors

In label-free optical biosensing tools, the bio-molecules are unlabeled or unmodified and are monitored in their natural form. There are different detection methods in label-free optical biosensors, including refractive index, optical absorption, and Raman spectroscopy. The utilization of refractive index changes, as the sensing transducer signal has been reported in various publications. Different optical label-free biosensors have been reported, such as photonic crystals, waveguides, ring resonators, fiber gratings, interferometers, and surface plasmon resonance [91]. Some of the refractive index biosensing platforms can monitor both the bulk solution refractive index (refractometer) and the refractive index variations stimulated by molecular bond formation (biosensor), while others can be employed only for the detection of the bulk solution refractive index. Both the refractive index and absorption are the real and imaginary measurement of the more general complicated refractive index constant and are correlated to each other by the Kronig–Kramers relationship. The Raman spectroscopy technique is unique because the target molecules are not labeled, as well as because the emitted Raman light is usually employed in biosensing, which is similar to the colorimetric/fluorescence-based label-driven technique [84,91]. Herein, we will limit the current study to colorimetric/fluorescence-based label-driven optical biosensors, as label-free optical biosensors have been extensively reported on in various review articles.

#### 4.1.2. Label-Driven Optical Biosensors

In cellulose optical biosensors, the type and quantity of the detected analyte can be evaluated by monitoring the UV-Vis spectral changes. The fundamental aspect in an optical biosensor depends mainly on finding chromatic substrate with a distinctive UV-Vis absorption (colorimetric or chromic biosensor) or emission (fluorescent biosensor) wavelength. The optical bio-sensing is one of the most well-known procedures in measuring a material concentration. In an optical biosensor, cellulose is utilized only as a supportive substrate for the incorporation of biological elements, such as enzymes, aptamers, and antibodies [92].

##### Cellulose-Based Fluorescent Biosensors

Cellulose can be usually activated before enzyme immobilization, and this could improve the biological element incorporation process in cellulose or increase the sensor efficiency. The utilization of a cross-linking agent is one of the most common cellulose activation methods. Cellulose can be activated by consecutive treatments with sodium periodate solution, ethylenediamine solution, and glutaraldehyde. Hence, different enzymes can be immobilized on the activated cellulose. Wang et al. [93] described the incorporation of cholesterol oxidase enzyme into a cellulose acetate membrane which had been previously activated by sodium periodate, ethylenediamine, and glutaraldehyde to introduce an optical fiber-based fluorescent biosensor. Yu et al. [94] described recently the preparation of a new fiber-optic glucose biosensor using glucose oxidase enzymes and fluorescent carbon quantum dots immobilized via dip-coating in a cellulose acetate film biosensor. The proposed optical fiber sensor showed a high sensitivity and reusability for the continuous online monitoring of low glucose concentrations. This allowed researchers to visualize real-time glucose fluctuations as a function of time. The varied emission intensity ratios displayed a linear relationship as a function of the glucose concentration in different ranges, including the micromole and nanomole levels, which was complied with the modified Stern–Volmer equation in the concentration range of 10–100 nmol L^−1^ with a limit of detection at 25.79 nM, and in the concentration range of 10–200 μmol L^−1^ with a limit of detection at 6.43 μM. The developed biosensor introduced a promising detection approach for traces of glucose in some complicated media, including the analysis of foodstuffs, biomedical purposes, and environmental monitoring. However, the quantitative analysis of trace glucose is still limited, owing to the small changes monitored in the fluorescence signal [94].

An et al. [95] reported recently the development of ratiometric fluorescent l-arginine and l-asparagine biosensors employing ethyl cellulose membrane, oxazine 170 perchlorate fluorophore, and enzymes immobilized in a matrix of ethyl cellulose membrane and polyurethane hydrogel. The sensory mechanism depended on the hydrolysis reaction of urea and l-arginine in presence of urease and arginase as catalysts. Thus, ammonia was generated in the presence of a l-arginine-based membrane. The hydrolysis reaction of l-asparagine was also detected under the l-asparaginase catalysis in the case of the l-asparaginase-based membrane. The ethyl cellulose membrane-based biosensor interacted with the generated ammonia to result in changes in the fluorescence intensity between 565 and 625 nm, at which the ratio of the fluorescence intensity was proportional to the concentrations of l-arginine and l-asparagine in the range of 0.1–10 mM. The biosensing membranes displayed a high quality in terms of the reversibility, stability, and response time. The interference study showed that some constituents, such as amino acids, had slight negative impacts on the efficiency of the biosensing membranes in detecting l-arginine and l-asparagine in both urine and blood samples as well as in the fermentation processes [95,96]. Schyrr et al. [97] investigated cellulose nanocrystals as an appropriate material for the simple integration of high-density biological sensors. Cellulose nanocrystals are characterized by their high porosity, large surface area, and high number of active surface functional groups. Porous cellulose nanocrystals and poly(vinyl alcohol) nanocomposite introduced fluorescent biosensor layers with a thickness in the range of 25–70 nm. The nanocomposite films were dip-coated on glass slides using an aqueous combination of cellulose nanocrystals and poly(vinyl alcohol), followed by heat curing to fix the porous nanostructure. Then, the hydroxyl surface functional groups partially interacted with 2-(acryloxy)ethyl (3-isocyanato-4-methylphenyl)carbamate to allow the integration of thiolated fluorescein-containing lysine active sites by a thiolene-based nucleophilic Michael addition. The produced biosensor films displayed a nearly immediate and obvious fluorescence intensity change responding to the pH variations. The technique was more extended toward the recognition of protease reactivity by incorporating Förster resonance energy transfer chromophoric pairs to the scaffold using a labile peptide sequence. The biosensing mechanism depended on degrading the protein linker in the presence of a suitable enzyme, leading to the chromophoric pairs separation and causing a “turn on” fluorescence which was originally quenched. Trypsin was monitored at a low concentration of 250 μg/mL, which is typical for monitoring wound fluids for abnormal proteolytic activity.

A cellulose acetate nanofibrous optical biosensor bearing an anionic fluorescent dendrimer was employed to detect metallo-proteins at low concentrations. The anionic fluorescent dendrimer was encapsulated in electrospun cellulose acetate nanofibers in the presence of deacetylated cellulose acetate to create a secondary porous architecture, which is desirable for improving the molecular interactions and consequently enhancing the sensing performance. The protein sensing performance of the produced fibers was studied by monitoring the quenching behavior of bovine serum albumin, cytochrome c, and hemoglobin as a function of the concentration. The quenching effect was attributed to the energy/electron transfer process among the iron-bearing protein and the fluorescent core [98]. Human neutrophil and porcine pancreatic elastases are serine proteases with a destructive proteolysis. They have essential functions against infection and digestion, respectively. Building on their earlier research work [99], Edwards and co-workers [100,101,102] were the first group to study the development of cellulose-based human neutrophil elastase biosensor by covalently bonding commercially available fluorescent peptide substrates onto a cotton cellulose nanocrystalline surface. Those fluorescent peptide biomolecules are characterized by their affinity and selectivity towards human neutrophil elastase existing in chronic wound fluids [103]. Human neutrophil/pancreatic elastases of glycine-esterified fluorescent tri- and tetrapeptides were covalently connected to cotton cellulose nanocrystals, as displayed in Figure 4. The degree of substitution of peptide immobilized on cotton cellulose nanocrystals was 3–4 peptides per 100 anhydroglucose units. Both the glycine and peptide/cellulose/nanocrystals displayed crystallinity index values at 79% and 76%, respectively, with a 58.5 Å crystallite size. The tripeptide conjugate model had a five-fold higher efficiency in the human neutrophil elastase than the tripeptide in solution. Additionally, the ability of the produced biosensor to monitor human neutrophil elastase was explored by applying leaving groups, including 7-amino-4-methylcoumarin and 4-nitroaniline. The 7-amino-4-methylcoumarin fluorophore demonstrated a higher sensitivity than 4-nitroaniline. The detection limits for 2 mg of both tri- and tetrapeptide cotton cellulose nanocrystalline conjugates over a 10-min reaction time course were monitored at 0.03 U/mL (porcine pancreatic elastase) and 0.05 U/mL (human neutrophil elastase), respectively. The prepared peptide-cellulose conjugate demonstrated a robust and sensitive performance with potential as a diagnostic device for elastase enzyme, and has been employed as a biomarker for a number of inflammatory diseases. However, potential drawbacks were monitored for the biosensors reported by Edwards and co-workers upon the direct detection of elastase enzyme in chronic wounds [101,102,103,104,105]. These drawbacks were related to the hydrolysis of the aromatic 7-amino-4-methylcoumarin and 4-nitroaniline from the surface of the cellulose. Firstly, there is a considerable concern about the potential toxicity to human cells, which may contribute to inflammation upon releasing those aromatic fluorophores into the wound tissue. Moreover, the continuous release of the fluorophore from the cellulose-based biosensor will lead to losing the signal over time. Those drawbacks have been arisen for many other dip-stick enzyme testing tools. Thus, there is a considerable scope to develop novel alternative strategies using bioactive cellulose-based enzyme biosensors that defeat those limitations. This will be helpful to advance the applications of cellulose biosensors for enzymes such as human neutrophil elastase.

Inspired by the previous research work of Edwards et al., Brumer and co-workers developed recently an alternate strategy toward cellulose-based esterase enzyme biosensors that overcomes issues related to fluorophore diffusion after substrate cleavage [105]. In this approach, the position of the fluorophore and peptide are switched with respect to the cellulose-linking moiety. Hence, the fluorophore remains attached to the cellulose surface, while the (non-chromophoric) biomolecular recognition element (peptide, lipid, or carbohydrate) is allowed to diffuse away to be ultimately metabolized (Figure 5).

To further enhance their technique, Brumer et al. [106] developed a chemo-enzymatic method to introduce more active sites by creating a clickable cellulose surface with the potential ability to attach a variety of functional groups via Cu(I)-catalyzed azide-alkyne 1–3 dipolar cycloaddition (Figure 6). The galactose oxidase-mediated oxidation of galactose was performed in the presence of catalase and peroxidase enzymes. This was followed by the reductive amination process of the produced galactose oxidase-oxidized polysaccharide using NaCNBH_3_ to introduce more reactive active sites per polysaccharide molecular unit. The alkyne-terminated xyloglucan was then adsorbed onto paper surface to offer a complementary Cu(I)-catalyzed clickable cellulose surface with 6-carboxyfluorescein tetraethylene glycol azide. Both the enzyme and time-dependent increments in fluorescence by the fluorescein fluorophore were monitored using a fluorescence scanner for a quantitative analysis and epi-illumination with a forensic flashlight for a naked-eye detection analysis. The produced alkynylated brush units provides a new method for the multifunctionalization of the cellulose surface for more adapted cellulose-based biosensors for other potential enzymes, such as human neutrophil elastase.

##### Cellulose-Based Colorimetric Biosensors

One of the main outcomes resultant from chronic kidney failure is the considerably increased concentration levels of metabolic waste products in blood, particularly urea, which is naturally discharged by the kidneys [107,108]. Thus, there have been various methods developed for the detection of urea, such as isotope-dilution gas chromatography mass spectrometry and electrochemical approaches [109,110,111]. However, those techniques are characterized as being time-consuming and complicated as well as having a poor specificity and sensitivity. Moreover, they miss the real-time live testing of the urea level, making them inappropriate for in vivo detection. Thus, there has been increasing demands for a simple and cost-effective diagnostic tool for clinical practice. In this context, Khattab et al. [112] studied recently the co-encapsulation of urease and molecular switching of tricyanofuran-hydrazone as active receptor sites in calcium cross-linked alginate microcapsules immobilized on a cotton gauze platform to function as a novel colorimetric biosensor for the naked-eye detection of urea [CO(NH_2_)_2_], as shown in Figure 7. The detection limit of urea was achieved as low as ~0.1 ppm under ambient conditions. The biosensor responded linearly depending on the total content of urea in the range of 0.1 and 250 ppm. The performance of the biochromic cellulose gauze biosensor depended mainly on the acid/base characteristic effect of the tricyanofuran hydrazone molecular switch.

The protonated form of the tricyanofuran hydrazone active receptor sites integrated within the calcium alginate microcapsules (1.7–12.9 μm) displayed a color change from light yellow to purple upon binding to ammonia generated from a hydrolytic urease/urea reaction. This can be attributed to the formation of the deprotonated anion form of the tricyanofuran-hydrazone, which exhibits a higher absorption wavelength compared to the protonated tricyanofuran-hydrazone form. The produced cotton gauze biosensor was studied using various analytical techniques, including scanning electron microscopy (SEM), energy-dispersive X-ray spectroscopy (EDX), and colorimetric measurements, as well as a Fourier-transform infrared spectroscopic (FT-IR) analysis. This approach can be utilized to determine the total content of urea in biological fluids, including the blood and urine, which may pave the way for the assembly of an implantable chromogenic smart bandage [112,113,114]. A similar study for the detection of urea was also reported by Khattab et al. [115] via introducing a colorimetric nanocomposite biosensor, which was prepared by immobilizing the tricyanofuran-hydrazone spectroscopic probe and urease enzyme in a matrix of calcium alginate and cellulose nanowhiskers (CNW) incorporated on cellulose strips. The detection limit was monitored in the range of 50–1100 ppm.

Li et al. [116] studied 1-ethyl-3-(3-dimethylaminopropyl)-carbodiimide/N-hydroxysuccinimide as a cross-linking agent in the development of a porous cellulose microsphere-based biosensor for glucose monitoring using “signal and color” outputs. Firstly, the plasma technology was applied to introduce active carboxyl groups to the cellulose matrix. Then, the glucose oxidase enzyme was chemically incorporated into the activated cellulose, using 1-ethyl-3-(3-dimethylaminopropyl)-carbodiimide/N-hydroxysuccinimide as a cross-linking agent. The prepared cellulose microgel demonstrated a rapid response within 4 min to glucose at 0.003 M. The response time was found to decrease to only 2 min upon increasing the glucose concentration to 0.005 M as the reaction solution displayed a color change from colorless to yellow. Imamura et al. [117] reported recently an oxidation process of a cellulose paper strip, using periodate to introduce an instrument-free colorimetric biosensor for protein quantification in urine with the ability to monitor the levels of proteinuria and microalbuminuria. The improvement of protein incorporation into the oxidized paper was quantified by a Ponceau chromophore-based colorimetric assay. The quantification depended on capturing the proteins via covalent bonding created with the carbonyl functional groups on the surface of the oxidized paper, followed by staining the area with Ponceau chromophore. A linear relationship was monitored amid the human serum albumin concentration and the length of the stained blot in the range of 0.1–3 mg/mL. This approach displayed a high sensitivity, with a quantification limit of 0.1 mg/mL. Paper-based biosensors are characterized by their low cost, lightness, flexibility, low volume, portability, and the possibility of chemical and physical modifications.

Colorimetric sensors have been also designed for the naked-eye detection of uranyl. A biosensor cellulose strip was fabricated by incorporating 2-(5-bromo-2-pyridylazo)-5-(diethylamino)phenol, as a chelating and chromogenic agent, onto electrospun cellulose acetate nanofibers. A color change was monitored from yellow to purple due to phase transition in presence of uranyl with a detection limit of 50 ppm [118,119]. A facile, rapid, and highly sensitive cellulose-based colorimetric enzyme assay was developed for the detection of glucose at low concentrations. This efficient biosensor was prepared via the immobilization of horseradish peroxidase (HRP) and glucose oxidase (GOx) enzymes onto cellulose previously oxidized with sodium periodate (Figure 8). When an aqueous solution of glucose was dropped onto the strip biosensor followed by adding 3,3′,5,5′-tetramethylbenzidine chromophore, a color change was monitored within only 5 min from colorless to blue, with a detection limit of 0.45 mM. Moreover, those strips have also been effectively employed for determining the glucose in actual urine samples. The color shift was simply reported using a camera to accomplish the semi-quantitative determination of glucose [120]. The morphological properties of the prepared biosensor were characterized by scanning electron microscopy (SEM), as shown in Figure 9. A unique three dimensional, network-like, and highly crowded porous structure of cellulose membrane (Figure 9a) was generated due to the phase separation of cellulose solution during the regeneration process. Figure 9b displayed the surface morphology of the oxidized cellulose membrane after chemical modification using sodium periodate oxidative process while maintaining the produced porous structure. However, the pore diameter was found to shrink, which was attributed to partial oxidative destruction of the six-membered ring structure of cellulose. The C-C single bond among C_2_ and C_3_ carbons of the cellulosic units was then opened, generating two aldehyde functional groups which could improve the intermolecular H-bond formation of the oxidized cellulose membrane. The aldehyde groups were then reacted with the primary amine substituents on glucose oxidase and horseradish peroxidase to introduce a Schiff-base bond. The cellulose-based strips showed a different morphology compared to that of the cellulose membrane and the oxidized cellulose membrane, as shown in Figure 9c. It displayed a network-like structure of lower porosity, which was attributed to the formed Schiff-base bond. Optical images of cellulose membranes prepared under different conditions are displayed in Figure 9d (wet), e (wet), and f (lyophilized). Compared to filter paper, the cellulose membranes demonstrated colorless and transparent properties under wet conditions. The elemental composition was also investigated using energy-dispersive X-ray spectroscopy (EDX), as shown in Figure 9a–c. The introduction of enzymes was proved by the presence of sulfur and nitrogen elements on the surface of the cellulose-based strips [120].

### 4.2. Cellulose-bBased Electrochemical Sensors

The amperometric sensors depend on the current that produced when a potential is applied between two electrodes (Figure 10). This current is related to the concentration of the analyte present in the system. This type of biosensor usually requires a transducer, as cellulose cannot be the only support for an amperometric sensor. Thus, the modification of cellulose for the immobilization of enzymes is an important role in amperometric sensors [121,122].

Saeed et al. [123] developed recently a non-invasive biosensor for human salivary glucose using a dialdehyde derivative of cellulose nanocrystals (CNCs) immobilized with gold nanoparticles (AuNPs) via an aminothiophenol crosslinker and glucose oxidase (GOx) enzymes via thioctic acid as a double crosslinker. The gold nanoparticles were attached to the cellulose via thiol chemistry. The monitored salivary glucose was compared to the serum glucose of the same individual. It possessed a high selectivity to glucose in the presence of common glucose interferences, such as uric acid, ascorbic acid, and dopamine. This biosensor displayed a good performance, with the ability to be applied for a real sample analysis. Mahadeva et al. [124] reported the immobilization of glucose oxidase onto cellulose/tin oxide (SnO_2_) hybrid nanocomposite toward the development of a low-cost and disposable glucose biosensor.

Atmospheric molecular oxygen was firstly physisorbed onto the biosensor surface active sites. Then, it was ionized via an electron extraction from the conduction band while shifting from one site to another. Thus, it was ionosorbed on biosensor surface active sites, such as (O^−^_ads_). Upon exposing the glucose oxidase previously immobilized onto a cellulose/SnO_2_ hybrid nanocomposite to glucose, an enzymatic reaction occurred amid the glucose oxidase and glucose (Figure 11). This leads to the generation of a D-gluconate-bearing H^+^ ion which interacts with (O^−^_ads_) and thereby releases the trapped electrons back to the tin oxide conduction band. The energy liberated during the decomposition of adsorbed molecules was found to be adequate for electrons to jump up into the conduction band to result in increasing the biosensor conductivity [125]. Due to the simple preparation process and facile accessibility of cellulose acetate, it has been considerably applied for medical diagnostic kits. Accordingly, the cellulose acetate-palmitate membrane has been employed as one of the best membranes for glucose/enzyme electrodes [126]. Urease enzyme was immobilized on a regenerated cellulose-tin oxide composite to function as a disposable and inexpensive transducer for urea monitoring, with a detection limit sensitivity of urea concentration as low as 0.5 mM [127]. When urease comes into contact with an aqueous solution of urea, NH_4_^+^ and CO_3_^−2^ ions are released, providing excess electrons to the tin oxide conduction band. This leads to improving the material’s conductivity.

Xiangling et al. developed a novel technique to fabricate an amperometric glucose biosensor by immobilizing the glucose oxidase enzyme onto gold nanorods (AuNRs) via a simple mixing process of glucose oxidase with AuNRs followed by cross-linking with a cellulose acetate medium using glutaraldehyde. The produced biosensor was found to exhibit a low detection limit, reproducibility, high sensitivity, and good storage stability [128]. A catalase-based biosensor was designed via the immobilization of the catalase enzyme onto cellulose acetate beads activated with cerium(IV) sulfate [Ce(SO_4_)_2_]. The immobilization of the catalase enzyme occurred via entrapment within activated cellulose acetate beads and cross-linking by glutaraldehyde. The activation of the cellulose acetate beads resulted in oxidizing its hydroxyl groups (OH) to aldehyde groups. Then, spacer arms were formed using bovine serum albumin [129]. Gilmartin et al. developed a uric acid sensor using both modified and unmodified cobalt phthalocyanine electrodes coated with cellulose acetate and loaded with uricase enzyme. The sensitivity was in the range of 1 × 10^−6^ to 13 × 10^−6^ mol/dm of uric acid, and the optimal uricase loading was 1 U [130]. Tsiafoulis and his workers developed an amperometric glycolic acid sensor based on glycolate oxidase/catalase immobilized in a cellulose acetate/polyvinyl acetate hybrid membrane. The immobilization of the enzyme was followed by superimposing with an outer polycarbonate membrane to protect the enzyme from leaking and microbial attack. This sensor showed a linear correlation between the biosensor sensitivity and the glycolate concentration level, which was in the range 0.01–1 mM, as the detection limit of glycolate was monitored at 6 μM [131]. A methyldopa biosensor was prepared via the immobilization of laccase from Aspergillus oryzae onto activated cellulose acetate mixed with ionic liquids, such as 1-butyl-3-methylimidazolium bis(trifluoromethylsulfonyl)imide. The squarewave voltammetric evaluation of the methyldopa biosensor showed a good linearity in the range of 34.8–370.3 μM and a detection limit of about 5.5 μM. It displayed reusability, good anti-interference, and reproducibility, as well as a satisfactory stability of about 60 days. Furthermore, it displayed a good level of accuracy for the detection of methyldopa in pharmaceutical samples. The preparation process of the laccase-based biosensor was performed, starting from cellulose acetate and 1-butyl-3-methylimidazolium bis(trifluoromethylsulfonyl)imide as an ionic liquid. In the presence of laccase as a catalyst, the methyldopa was oxidized to quinone, which was electrochemically reduced on the biosensor surface back to methyldopa at a potential of +0.07 V vs. Ag/AgCl, as shown in Figure 12 [132].

In addition, the laccase-based biosensor was fabricated to monitor catechol, employing the laccase enzyme immobilized onto graphene/cellulose microfibrous composite, which was applied onto screen-printed carbon electrode. The response time, sensitivity, and detection limit values were 2 s, 0.932 μM μA^−1^ cm^−2^, and 0.085 μM, respectively. Therefore, this type of biosensor can be used for redox active proteins [133]. Glucose oxidase enzyme was immobilized on a fine carbon powder followed by cross-linking with a cellulose acetate membrane, which was coated with hydrogel layers bearing positive and negative charges. This new enzyme electrode exhibits a lifetime of three months, after which it can be recharged with a fresh oxidase. It displayed a stable and linear responsiveness to glucose at concentration levels around 5300 mg/dL in the presence of either glucose only or with other interferences, including bilirubin, urea, creatinine, ascorbic acid, L-cysdne, glycine, and uric acid [134]. Tsiafoulis et al. developed a simple and cost-effective approach for the detection of glycolic acid in real samples. A mixture of glycolate oxidase/catalase enzymes was incorporated into a sandwich membrane through physical adsorption. The sandwich membrane consisted of polycarbonate as an outer membrane and cellulose acetate as an inner membrane. The membrane assembly was mounted on an amperometric flow-cell or on an oxygen electrode [131,135]. The Tkác et al. work research group designed a voltammetric fructose biosensor by immobilizing fructose dehydrogenase on ferrocene and ferrocene/Nafion-modified cellulose acetate membrane. The initial sensitivity of the electrodes was about 226 nA/mM. Both the ferrocene and ferrocene/Nafion-modified cellulose acetate-based membranes demonstrated a high stability [136].

The 8-hydroxy-2′-deoxyguanosine compound has been associated with debilitating diseases, such as autism and Alzheimer disease. Thus, Sales’ group developed a biosensor capable of monitoring 8-hydroxy-2′-deoxyguanosine generated in DNA under oxidative stress conditions. This biosensor was prepared from cellulose paper strips pre-coated with a conductive carbon-based ink to introduce a catalytically active biosensor electrode [137]. The porous cellulose paper strips have been applied as rigid or semi-rigid scaffolds. The porosity enables storing the immobilized reagents in dry-state for utilization in the analysis of a liquid sample. Cinti et al. synthesized Prussian Blue Nanoparticles immobilized onto filter paper (Paper Blue) without forming wastes and without external inputs, such as reducing agent, pH, and voltage. Glucose oxidase immobilized onto Paper Blue was employed for the detection of blood glucose using electrochemical biosensors for a wide range of concentrations up to 25 mM (450 mg/dL) [138]. Xiaosong and his co-workers developed a paper-supported aptasensor, which is a certain class of biosensor where the biological active detection component is either DNA or RNA aptamer. This paper-supported aptasensor depended on luminescence resonance energy transfer from the up-conversion nanoparticles to AuNRs for the accessible detection of exosomes. In the presence of exosomes, the two segments of the aptamer can merge with the surface protein of the exosomes, forming a conjugation to close the distance between the up-converted nanoparticles and the gold nanorods. This can then initiate the luminescence resonance energy transfer and stimulate the luminescence quenching. These changes can be monitored by a homemade imaging system reaching a detection limit of exosomes as low as 1.1 × 10^3^ particles/μL [139]. The incorporation process of acetylcholinesterase into the membrane of cellulose acetate and glutaraldehyde as a working electrode with Ag/AgCl as a standard electrode was developed as a biosensor for diazinon pesticide with a detection limit of 10^−6^ ppm and a sensitivity of around 20.275 mV/decade [140].

Pandey et al. [141] presented recently a potentiometric biosensor for cholesterol using cholesterol oxidase enzymes immobilized on single-walled carbon nanotubes, which in turn were loaded in a cellulose acetate membrane. The electrocatalytic responsiveness of the cholesterol oxidase/carbon nanotubes/cellulose acetate biosensor to detect cholesterol showed a better performance compared to the cholesterol oxidase/cellulose acetate biosensor. It showed a high sensitivity with a detection limit of 10^–8^ M and a linear relationship in the range of 10^–3^–10^–8^ M. Alpat et al. [142] introduced a new voltammetric biosensor for ethanol employing alcohol dehydrogenase, which was co-immobilized with the reduced form of nicotinamide adenine dinucleotide (NADH) on the surface of a glassy carbon electrode adjusted with cellulose acetate bonded to tolonium chloride, known as a blue cationic (basic) dye. The surface was then coated with glutaraldehyde/bovine serum albumin as a cross-linking agent (Figure 13). The produced biosensor demonstrated a high sensitivity and selectivity to ethanol, with a detection limit of 5.0 × 10^−6^ M and a linear responsiveness in the range of 1 × 10^−5^ M and 4 × 10^−4^ M. However, the biosensor displayed 50% of its preliminary activity after storage for 20 days. The developed biosensor showed a good sensitivity, accuracy, and selectivity for the detection of ethanol, with a fast response and low detection limit. It also displayed a long-term storage stability and a good thermal stability. Maniruzzaman et al. [143] studied the fabrication of titanium dioxide(TiO_2_)/cellulose nanocomposite as a conductometric biosensor for glucose at low cost, flexibility, and disposability. The biosensor was prepared by dispersing the cotton pulp using *N*,*N*-dimethylacetamide/lithium chloride as a solvent, followed by adding the TiO_2_ nanoparticles to create the nanocomposite. The glucose oxidase enzyme was integrated into this nanocomposite by physical adsorption, leading to covalent bonding between the glucose oxidase and TiO_2_, which was proved by X-ray photoelectron spectroscopy. The linear responsiveness of the glucose biosensor was monitored in the range of 1–10 mM.

Another economical, flexible, and disposable conductometric biosensor for glucose was prepared from cellulose and zinc oxide (ZnO) composite film. Cotton pulp and ZnO nanoparticles (NPs) were dispersed in a mixture of *N*,*N*-dimethylacetamide (DMAc), lithium chloride (LiCl), and sodium dodecyl sulfate (SDS), followed with curing in a mixture of *iso*-propanol (IPA) and deionized (DI) water. Glucose oxidase was then immobilized in the produced nanocomposite via physical adsorption (Figure 14). The enzyme reactivity in the biosensor was found to increase upon increasing the weight ratio of the ZnO nanoparticles. This was attributed to the composite surface morphology and the increased crystallinity of ZnO in the composite film, which led to increasing the current level, consequently increasing the enzyme activity in the biosensor. The ZnO/cellulose composite film was capable of monitoring glucose in the concentration range of 1–12 mM [144]. To assess the electrochemical properties, a cell was assembled consisting of a gold wire with a diameter of 0.3 as one electrode and the ZnO/cellulose composite layer as the reference electrode. Sputtering gold onto both top sides of the ZnO/cellulose composite layer was carried out before immobilizing the glucose oxidase enzyme. Aqueous solutions of glucose were prepared at different concentrations (1–20 mM) to be employed as an electrolyte. The current slope vs. potential (I-V) curve (Δ*I*/Δ*V*) was applied as an indicator for the enzyme reactivity [144].

## 5. Conclusions and Future Perspectives

This review presents the usage of cellulose as a supporting material for biosensors in biomedical diagnostics. It was found that cellulose can be applied for a variety of promising biosensing purposes for the detection of various bio-molecules, such as urea, lactate, glucose, genes, amino acids, cholesterol, and proteins. Cellulose and its derivatives have unique chemical structures, as they provide a good platform to accomplish the immobilization process of bioactive molecules in biosensors. Cellulose-based biosensors are promising platforms that are characterized by being low-cost, portability, lightness, miniaturized, and consumer friendly, which meet the requirements for on-site detection. Cellulose and its derivatives have been shown as a versatile substrate with a distinctive molecular structure which introduces a high-quality bioassay for the development of novel bio-materials and bio-tools. The high number of hydroxyl functional groups accessible on the cellulose polymer chains resulted in a useful solid material with the ability to undergo chemical modifications, allowing the construction of novel materials for new advanced biosensor-based applications. We presented a summary of various types of cellulose matrices and various detected analytes as well as discussing the potential applications of those cellulose matrices as optical (colorimetric and fluorescent) and electrochemical biosensors. Due to striving to utilize low-cost and environmentally friendly materials, we believe that the demands for cellulose-based biosensors will considerably be increased. Thus, it is anticipated that various future research will be inspired to explore the beneficial outcomes of using cellulose-based biosensors for healthcare, biodefense, food-safety, and environmental monitoring, as well as for biological and medical applications. The immobilization of nanomaterials within the nano- or microstructure of cellulose brings novel strategies for improving their analytical efficiency. There are numerous cellulose-based biosensors that have been developed for a broad range of applications at the laboratory scale. However, only a few cellulosic biosensors are commercially available. Thus, more efforts are needed in research innovation toward the development of automated, real time, and continuous biosensing devices for healthcare and medical applications.

## Figures and Tables

**Figure 1 biosensors-10-00067-f001:**
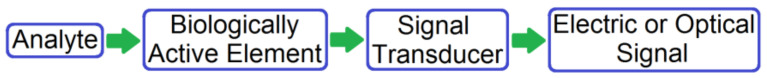
Schematic diagram representing the different components of a biosensor.

**Figure 2 biosensors-10-00067-f002:**
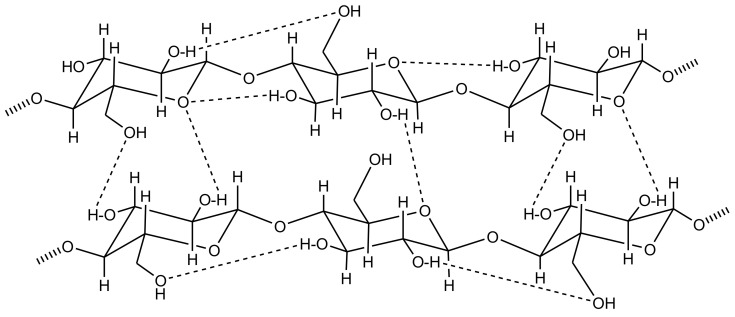
Cellulose structure demonstrating intra- and intermolecular hydrogen bonding patterns.

**Figure 3 biosensors-10-00067-f003:**
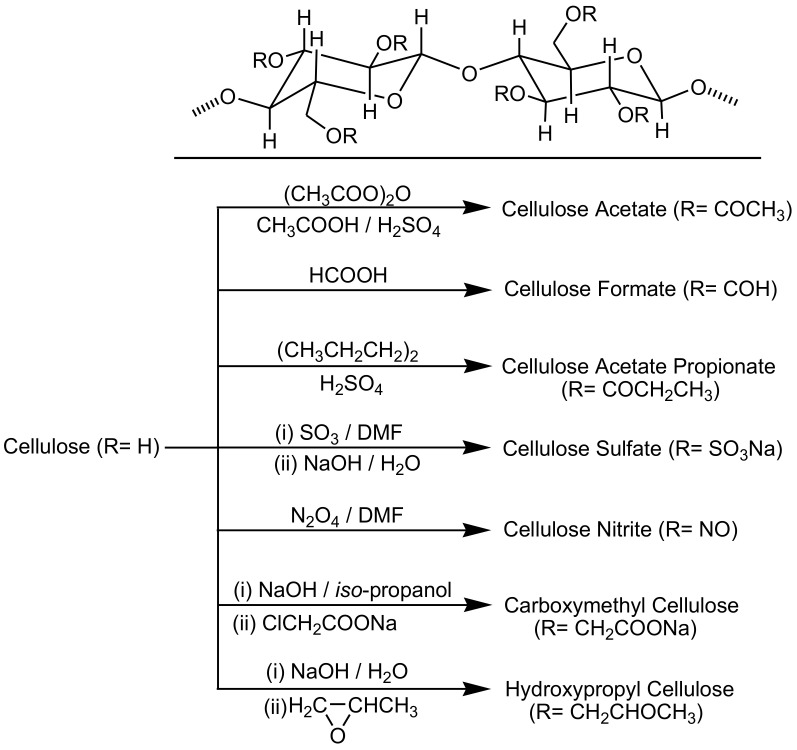
Preparation pathways of various cellulose derivatives.

**Figure 4 biosensors-10-00067-f004:**
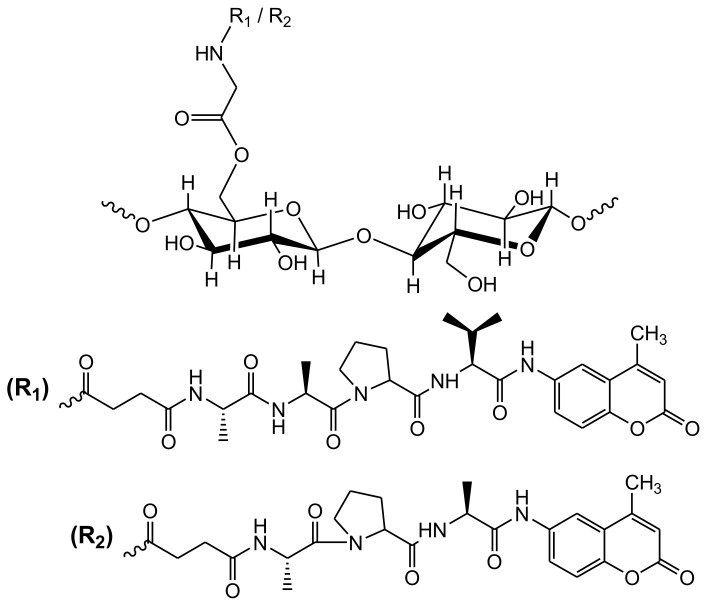
Structure of cotton cellulose nanocrystals-[O-C(O)Gly-NHC(O)]succinyl-Ala-Ala-Pro-Val-AMC (R_1_; tetrapeptide derivative) and cotton cellulose nanocrystals-[O-C(O)Gly-NHC(O)]succinyl-Ala-Pro-Ala-AMC (R_2_; tripeptide derivative).

**Figure 5 biosensors-10-00067-f005:**
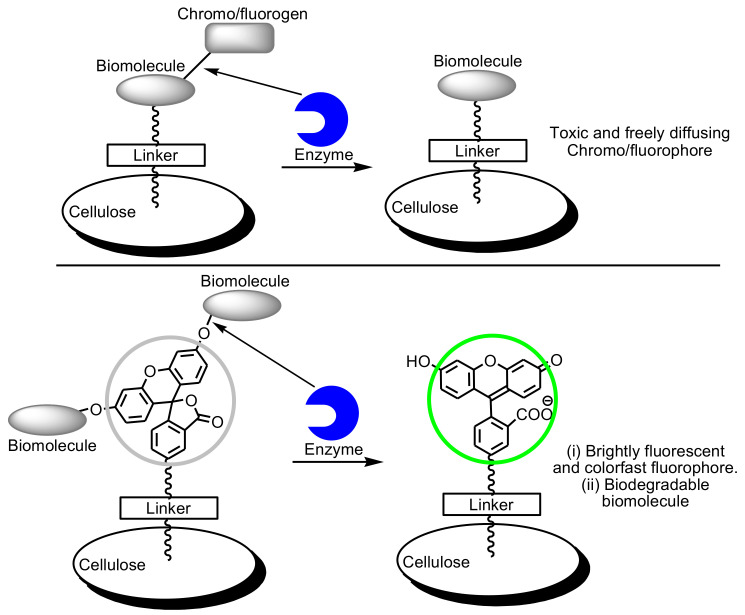
Cellulose-based enzyme biosensor comprising a surface linked with biomolecules (**top**), and a surface linked with fluorogenic moiety (**bottom**).

**Figure 6 biosensors-10-00067-f006:**
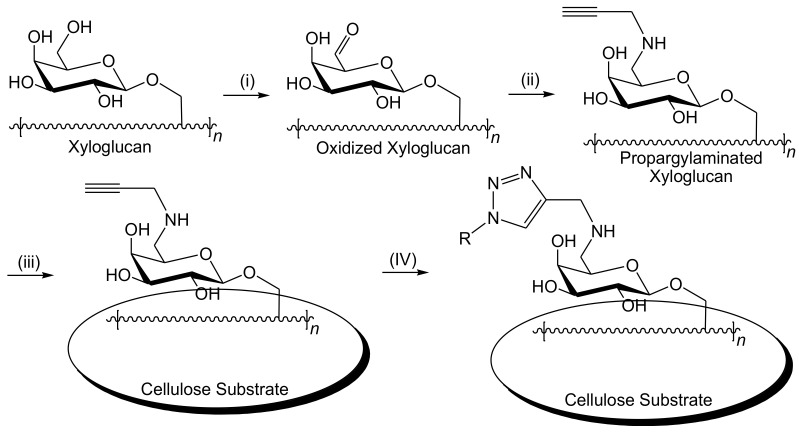
Chemo-enzymatic development of the cellulose-based esterase biosensor via the activation of cellulose surface using galactosylated hetero-polysaccharides; (**i**) galactose oxidase-mediated oxidation using O_2_, galactose oxidase, catalase, and peroxidase at room temperature in H_2_O; (**ii**) reductive amination by propargylamine, NaCNBH_3_, and CH_3_COOH at room temperature in H_2_O or H_2_O/CH_3_OH (2:1); (**iii**) physical adsorption onto cellulose substrate at room temperature in aqueous medium; (**iv**) in situ Cu(I)-catalyzed click reaction by applying 6-carboxyfluorescein tetraethylene glycol azide (RN_3_), CuSO_4_, sodium ascorbate, and tris-hydroxypropyl triazolylamine in a mixture of H_2_O/butanol (7:3).

**Figure 7 biosensors-10-00067-f007:**
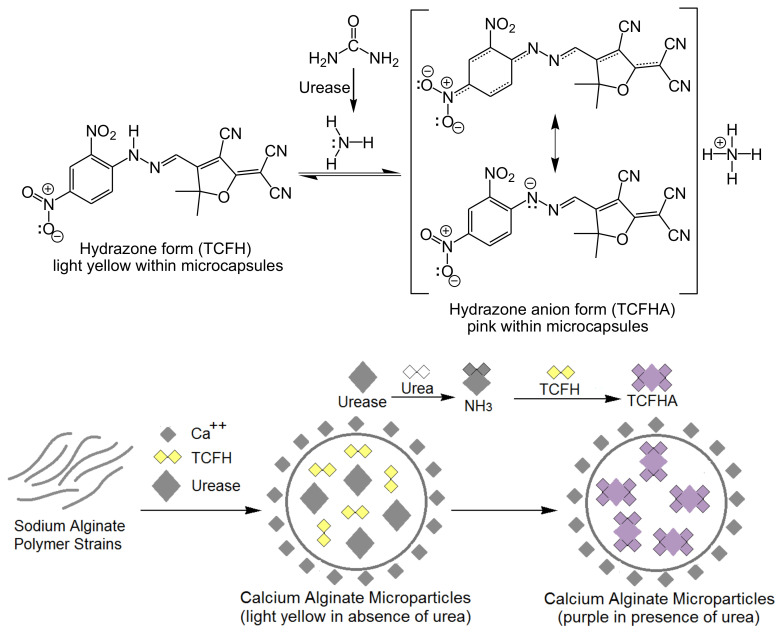
Proposed mechanism (**top**) and schematic diagram (**bottom**) for the colorimetric detection of urea using tricyanofuran-hydrazone active receptor sites; TCFH is tricyanofuran-hydrazone, TCFHA is tricyanofuran-hydrazone anion. “Reprinted with permission from Elsevier [112]”.

**Figure 8 biosensors-10-00067-f008:**
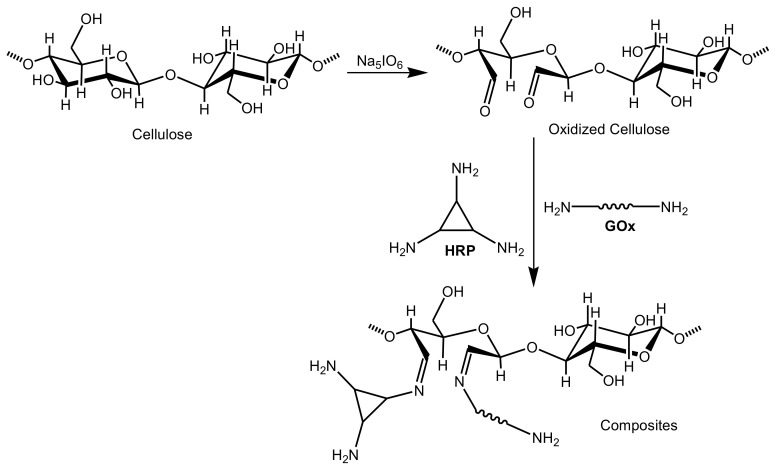
Preparation of cellulose strips for the determination of glucose.

**Figure 9 biosensors-10-00067-f009:**
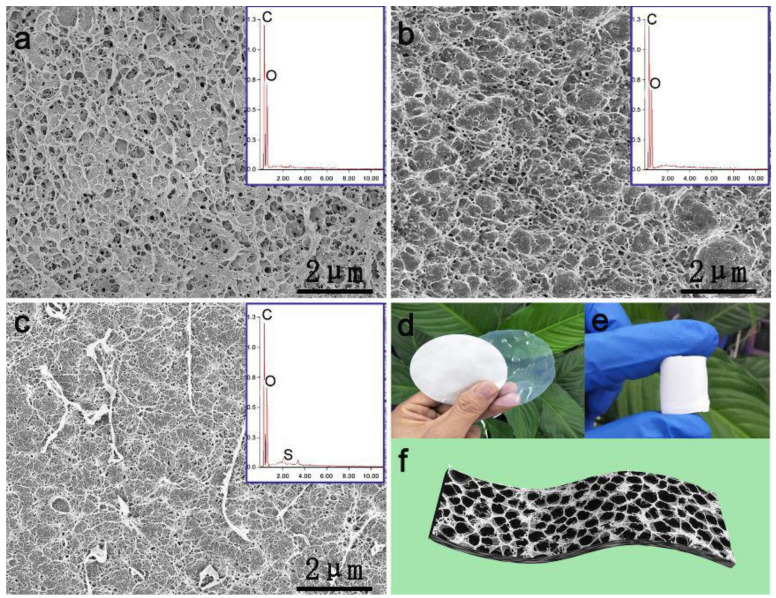
SEM images and inset X-ray spectroscopy (EDX) diagram of (**a**) cellulose membrane, (**b**) oxidized cellulose membrane, and (**c**) cellulose-based strips; optical images of (**d**) wet and (**e**) lyophilized cellulose membranes; and (**f**) microscopic schematic diagram of cellulose membrane. “Reprinted with permission from American Chemical Society [120]”.

**Figure 10 biosensors-10-00067-f010:**
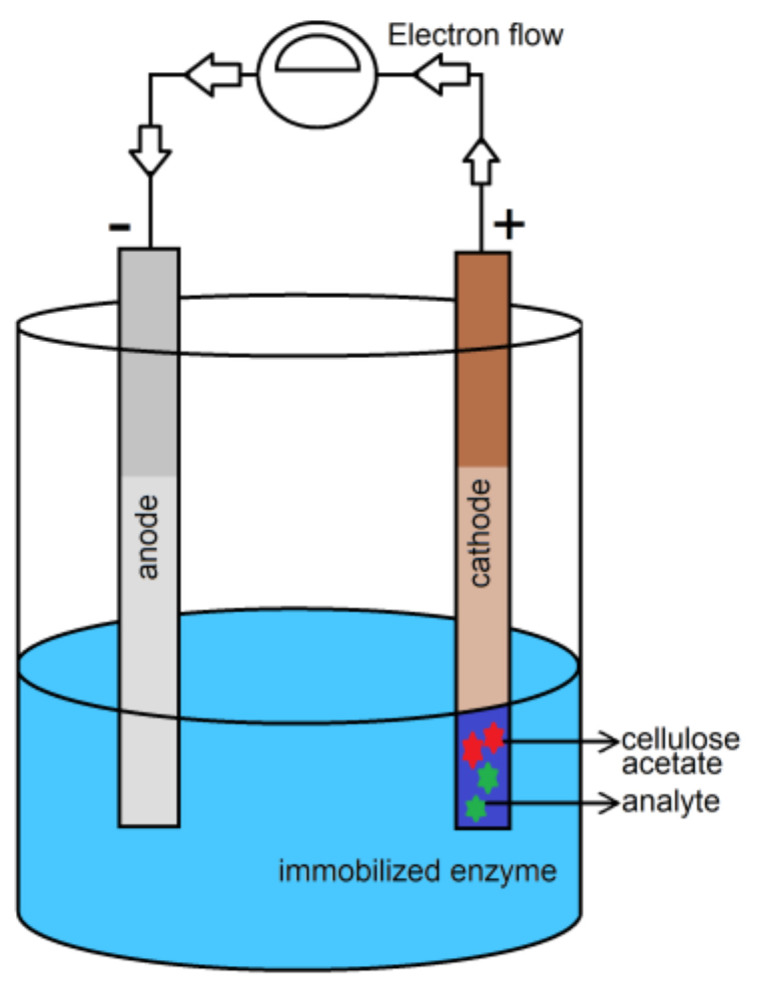
Schematic diagram demonstrating a biosensor testing for the amperometric determination of an analyte based on cellulose immobilized onto the cathode.

**Figure 11 biosensors-10-00067-f011:**
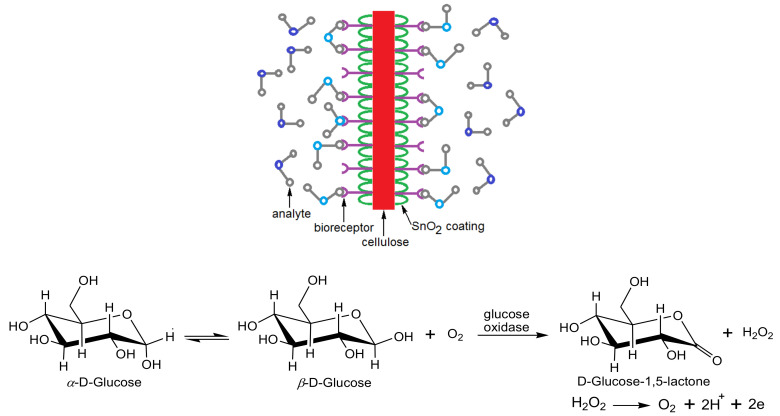
Conductometric glucose biosensor (top) and the sensing reaction mechanism (bottom).

**Figure 12 biosensors-10-00067-f012:**
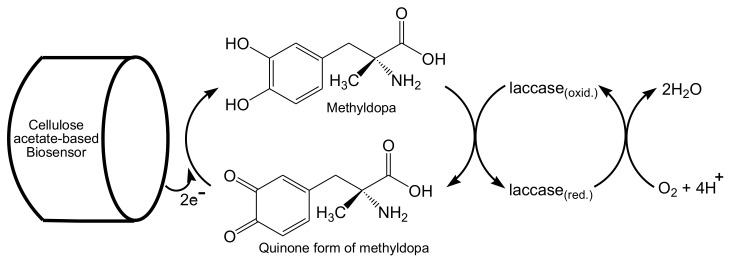
Schematic diagram representing the laccase-catalyzed oxidation of methyldopa with a sequent electrochemical reduction process.

**Figure 13 biosensors-10-00067-f013:**
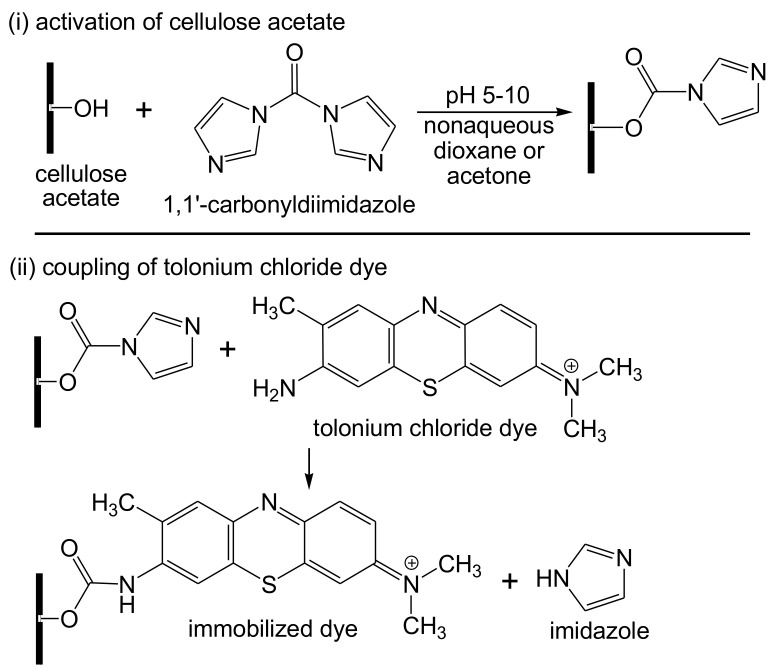
Immobilization of tolonium chloride (toluidine blue O) dye onto cellulose acetate.

**Figure 14 biosensors-10-00067-f014:**
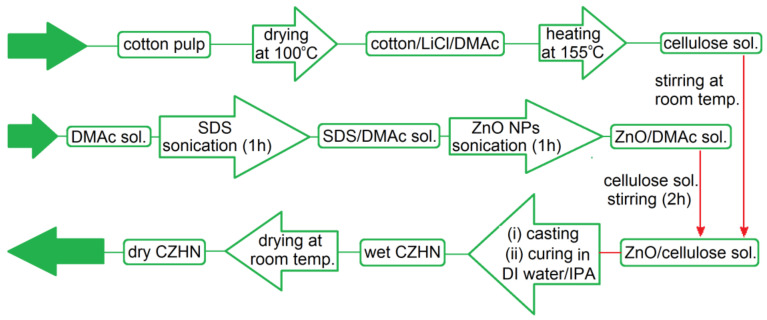
Fabrication procedure of a glucose oxidase-immobilized ZnO/cellulose composite film biosensor.

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
