# Peer review of "Recent Advances in Cellulose-Based Biosensors for Medical Diagnosis"

_biosensors, 2020, doi:10.3390/bios10060067_

Round 1

Reviewer 1 Report

This paper is a review of the published literature on the application of cellulose material for creating create a variety of biosensors.  The advantage of cellulose and its derivatives is that the unique chemical structure provides a good platform to immobilize bioactive molecules.  The review summarized different types of cellulose matrices and detected analytes for creating optical (colorimetric and fluorescent) and electro-chemical biosensors.  In general, the topic is very interesting and the material presented is technically sound.

However, the main issue with the current manuscript is that it is written as a collection of illustrative examples and loosely connected ideas with little attempt to integrate the information or provide unique insight into the topic.  This is evident in both the very generic abstract and Introduction where the authors do not explicitly state what aspects of cellulose materials are beneficial or essential for realizing low-cost biosensors. Stating that it has “distinctive chemical structure” means very little without proper context or explanation. 

From the readers’ perspective, key questions about the topic concern why and how cellulose and its derivatives improve sensitivity, accuracy, and rapid response.  Section 2 is an overview of  the components of a typical biosensor but does not provide any insight into the specifics of the role of the cellulose material.  Finally, the examples are largely summaries taken from work reported in the published literature with little discussion about broader issues of using cellulose materials for fabricating functional biosensors.

Author Response

Reviewer # 1:

This paper is a review of the published literature on the application of cellulose material for creating create a variety of biosensors.  The advantage of cellulose and its derivatives is that the unique chemical structure provides a good platform to immobilize bioactive molecules.  The review summarized different types of cellulose matrices and detected analytes for creating optical (colorimetric and fluorescent) and electro-chemical biosensors.  In general, the topic is very interesting and the material presented is technically sound.

- However, the main issue with the current manuscript is that it is written as a collection of illustrative examples and loosely connected ideas with little attempt to integrate the information or provide unique insight into the topic.  This is evident in both the very generic abstract and Introduction where the authors do not explicitly state what aspects of cellulose materials are beneficial or essential for realizing low-cost biosensors. Stating that it has “distinctive chemical structure” means very little without proper context or explanation. 

Authors Response: Thanks for the reviewer comment. The manuscript was fully revised to strengthen the connection between different topics.

- From the readers’ perspective, key questions about the topic concern why and how cellulose and its derivatives improve sensitivity, accuracy, and rapid response. Section 2 is an overview of  the components of a typical biosensor but does not provide any insight into the specifics of the role of the cellulose material.  Finally, the examples are largely summaries taken from work reported in the published literature with little discussion about broader issues of using cellulose materials for fabricating functional biosensors.

Authors Response: The reasons for the improved sensitivity, accuracy, and rapid response mainly depend on the natural fibrous, high surface area and porous structure of cellulose and its derivatives. This large surface area and porous structure of a fibrous cellulose substrate result in rapid adsorption and diffusion of the analyte to the active detective sites through the mesh. This was highlighted in manuscript for clarification. However, we did not repeat those properties at different positions in manuscript to avoid repetition. In this review, we presented a summary of various types of cellulose matrices and various detected analytes as well as discussing the potential applications of those cellulose matrices as optical (colorimetric and fluorescent) and electrochemical biosensors. The major idea was to discuss the structure and modifications of the cellulose polymer chains s allowing the construction of novel materials for new advanced biosensing applications.

Reviewer 2 Report

Kamel et al. reported a review paper on the summarization of the recent developments of cellulose-based biosensors. The topic is interesting for the reader of this journal. However, some revisions have to be done before it is accepted for publication.

  • The title is a bit confusing. The main idea of this review paper, for my understanding, is to summarize the advances on cellulose-based biosensor. I suggest to change the title accordingly.
  • The organization of this review paper is not so clear and logic. I suggest to re-devide the structure according to the new title.
  • The typical figures and schemes to show the sensing mechanism are lacking and not so impressive.
  • The challenges in cellulose based biosensors construction is not clearly presented, for examples, how to do the surface modification and blocking.

Author Response

Reviewer # 2:

Kamel et al. reported a review paper on the summarization of the recent developments of cellulose-based biosensors. The topic is interesting for the reader of this journal. However, some revisions have to be done before it is accepted for publication.

- The title is a bit confusing. The main idea of this review paper, for my understanding, is to summarize the advances on cellulose-based biosensor. I suggest to change the title accordingly.

Authors Response: Thanks for the reviewer recommendation. The title was changed to “Recent Advances in Cellulose-based Biosensors for Medical Diagnosis”.

- The organization of this review paper is not so clear and logic. I suggest to re-devide the structure according to the new title.

Authors Response: Thanks for the reviewer recommendation. The manuscript was fully revised to strengthen the connection between different topics. However, the best organization of the review should depend on the measurable signal. The recognition/detection mechanisms of the cellulose-based biosensors demonstrated two major classes of measurable signal generation, including optical and electrochemical cellulosic biosensors. As a result of their simplicity, high sensitivity and low cost, cellulose-based optical biosensors were particularly of great interest to include label-free and label-driven (fluorescent and colorimetric) biosensors. There are many types of cellulose and cellulose derivates as well as enzymes. Thus, it will be very difficult to re-divide the manuscript according to different types of enzymes or different types of cellulose structures.

- The typical figures and schemes to show the sensing mechanism are lacking and not so impressive.

Authors Response: Thanks for the reviewer recommendation. The manuscript was fully revised to improve figures and schemes. Also, more significant schemes and figures were added to enrich the manuscript.

- The challenges in cellulose based biosensors construction is not clearly presented, for examples, how to do the surface modification and blocking.

Authors Response: Thanks for the reviewer recommendation. We discussed further the challenges in cellulose based biosensors construction as well as cellulose surface activation/modification.

Thank you very much!

Round 2

Reviewer 1 Report

The manuscript is a literature review of cellulose-based biosensors for medical applications. Cellulose and its derivatives provide a platform to immobilize bioactive molecules. In general, the authors have satisfactorily addressed many of the concerns raised in the previous review. 

Some of the main revisions to the original version include a new more focused title, several explanatory sentences in the Abstract that clarify the types of biosensor considered in the review, more details about elastase-specific biosensors, three new figures and related discussion, and 15 additional references to help establish context and support material in the review.

Although technically sound and acceptable as a review paper, it would have been a stronger manuscript if it provided unique insight(s) into cellulose-based biosensors rather than just summarizing different approaches.

Reviewer 2 Report

I think the review paper has been improved and ready for publication.